theoretical biology/biophysics/cellular biophysics

biothermodynamics, complex systems, biophysical resonance, extremely low-frequency electromagnetic field, cancer cell migration, cancer growth

**Author for correspondence:**
Francesca Silvagno
e-mail: francesca.silvagno@unito.it

# Thermomagnetic resonance affects cancer growth and motility

Umberto Lucia[1], Giulia Grisolia[1], Antonio Ponzetto[2], Loredana Bergandi[3] and Francesca Silvagno[3]

[1]Department of Energy, Politecnico di Torino, Corso Duca degli Abruzzi 24, 10129 Torino, Italy
[2]Department of Medical Sciences, University of Torino, Corso A.M. Dogliotti 14, 10126 Torino, Italy
[3]Department of Oncology, University of Torino, Via Santena 5 bis, 10126 Torino, Italy

  UL, 0000-0002-3123-2133; LB, 0000-0003-0951-7340; FS, 0000-0002-8800-9135

The fight against a multifaceted incurable disease such as cancer requires a multidisciplinary approach to overcome the multitude of molecular defects at its origin. Here, a new thermophysical biochemical approach has been suggested and associated with the use of electromagnetic fields to control the growth of cancer cells. In particular, thermodynamic analysis of the heat transfer is developed in correlation with cellular parameters such as the volume/area ratio. We propose that the electromagnetic wave, at the specific frequency calculated as the characteristic response time of any cell type to the external thermal perturbation, can affect resonant intracellular molecular oscillations. The biochemical model hypothesizes that microtubules are stabilized, and the impact is predicted on cell growth, migration and mitochondrial activity. Experimental validation of the theoretical results shows that the thermodynamic analysis allows the application of the specific electromagnetic field able to decrease cancer cell invasion and proliferation.

## 1. Introduction

From a thermodynamic point of view, a living system is an adaptive open system, thus non-equilibrium thermodynamics represents a very effective analytical tool [1].

The life of cells, including cancer cells, is a thermochemical process that converts energy into work. Cells live in an environment that slowly changes its chemical and physical properties. In mammals, organs and tissues are generally endowed with a homeostatic capability, which is characterized by

internal thermal regulation. In thermodynamics, wasted heat is considered non-organized information, therefore, the dissipated heat does not represent a form of communication between cells. Consequently, the effect of a perturbation imposed on the cells can be considered, initially, as the sum of the responses of all cells, causing an effect on the whole tissue. This principle can be applied to a perturbation producing heat waste, where the production of heat is the global response of the cell system [2].

This system, considered as single cells in a culture, is an open system; therefore, energy and matter flow through its border, while biochemical and biophysical transformations occur within it, with a related net production of entropy. The environment of the considered system is composed of the aqueous solution with cellular nutrients in suspension, along with the substances discarded by the cells and the gaseous atmosphere above the suspension. As a consequence, the biosystem is composed of the cell membrane (which delimits the volume of the cell and controls inflows and outflows of molecules), the cytoplasm and the organelles [3].

Systems and environments continuously change; in the environment, the concentrations of some molecular species decrease over time because nutrients must flow into the system in order to feed the biochemical reactions and produce macromolecular cell components, with a related increase of cell mass and volume [4]. These biochemical reactions require energy, with a related waste of heat towards the environment. The net effect of all these biochemical reactions is reducing the entropy of the system, and increasing the generation of entropy in the environment [5].

Analysing chemical species and their reactions in living cells has identified some biochemical sequences that start with nutrient molecules and end with the formation of organic products, wasted molecules and wasted heat. Characteristics of living systems have been determined by using batch cultures grown at their optimum temperature; however, this experimental method presents disadvantages because the living system is forced to live in a continuously changing environment, consuming nutrients and releasing waste products and waste heat. Cells exchange energy and matter through their membrane, in order to maintain their living conditions [2]. All biophysical and biochemical processes require fluxes of energy, ions and molecules, which are controlled by endogenous electric fields and accumulated in the nanometre-thin layer of water. Several processes are driven by fluxes of energy and mass, and cell shape is fundamental in their analysis [3]. For example, differentiated cells are hyperpolarized compared with quiescent or cycling cells, and hyperpolarization increases the efflux of some ions (e.g. $Ca^{2+}$, $K^+$, $Zn^{2+}$, etc.). In addition, cell motility requires energy and ion fluxes [6,7]. Direct cell migration is fundamental in tissue formation, but uncontrolled proliferation and invasion lead to new behaviours, and cancer emerges as a disease of abnormal growth. From a thermodynamic point of view, living systems are open, complex systems endowed with the control of fluxes.

Although the Gibbs free energy could seem the correct function for studying the steady states of living systems, its value as a criterion for establishing spontaneous evolution is, in fact, limited to the complex phenomena that occur at constant temperature and pressure inside the specific living system. A more general function is required to properly analyse living systems. Considering that the system wastes energy and mass generating irreversibility, it is conceivable that the general criterion for studying its spontaneous evolution can be the entropy generation related to the changes of the system. Entropy generation always increases in any spontaneous and irreversible evolution.

The aim of this study was to thoroughly explore the use of heat in the thermodynamic analysis of a cancer cell system. Recently, we have used the entropy generation to introduce a thermodynamic approach for analysing cancer cells [8]. Indeed, in our recent study, we have shown that the thermodynamic approach reveals a new perspective in analysing living systems. In fact, when we stimulated cancer cells with an extremely low-frequency electromagnetic field (ELF-EMF), we were able to select the frequency driving a global response evaluated as entropy variation. The present study aimed to further develop the thermodynamic formulation of this approach, by introducing thermal exchange analysis and by describing the resonant effects triggered by the exposure of cells to ELF-EMF.

# 2. Material and methods

## 2.1. Cell cultures and treatments

Two human cancer cell lines, representative of two different human cancers, were used in this study; MDA-MB-231 is a human triple-negative breast adenocarcinoma cell line, and HT-29 is a human

colon cancer cell line. Cell lines were obtained from the American Type Culture Collection (ATCC; Rockville, MD, USA). Cells were grown in Dulbecco's modified Eagle medium (DMEM) with 4.5 g l$^{-1}$ glucose medium containing 2 mM L-glutamine, 1% (v/v) antibiotics (penicillin/streptomycin solution) and 10% (v/v) fetal bovine serum (FBS) at 37°C in a humidified atmosphere at 5% $CO_2$. Cell size was evaluated by photographs taken in different areas of the dish. For each cell type, cellular and nuclear sizes were calculated using an ImageJ software analysis on 30 images (Sun Microsystems Inc., Palo Alto, CA). Values are reported in the electronic supplementary material.

During experiments, cells were seeded in 96-multiwell plates for proliferation assays, or six-multiwell plates for JC-1 and PCR analysis, and were continuously exposed to ELF-EMF for the required time, whereas control cells were grown in the same incubator under standard conditions.

## 2.2. ELF-EMF exposure system

The experimental set-up has been previously described [8,9]. Briefly, the exposure system was created by two independent pairs of coaxial coils wound into a frame of cylindrical shape, with an outer radius of 8 cm and a distance between the two coaxial coil couples of 8 cm. The cell culture dish was inserted into the central part of the apparatus and continuously exposed to ELF-EMF for the required time. The inner coils (7.5 cm radius) were connected to an AC current generator, producing a sine wave at an intensity of 100 µT and at the specific frequencies calculated by the mathematical model (6 Hz for MDA-MB-231 cells and 24 Hz for HT-29 cells). The outer coils were supplied with a DC current, which provided a constant magnetic field of 45 µT, which is the average value of the Earth's magnetic field. The experimental set-up was shielded from the background magnetic field by a box made of a 2 mm thick inner layer of mu-metal and an outer layer produced by a special aluminium-free alloy provided by the company G-Iron of Arezzo (Italy).

## 2.3. Wound healing assay

The assay was carried out as previously described [10]. Cells were seeded in a 24-well plate and exposed to the specific frequency of ELF-EMF for 2 days in growth medium; they were then starved overnight again under exposure, with the aim of stopping proliferation. Next, a scratch was generated in the monolayer with a sterile pipette tip, followed by the exposure to ELF-EMF for another 24 h. Control cells were processed in the same manner but in the absence of ELF-EMF. Images of wound closure were obtained at 0 and 24 h using a light microscope at 20× magnification with a digital camera under bright field illumination. The area of the wound was measured in the central part of each well by using ImageJ software. Measurements were then converted into a percentage of wound closure using the equation: $100 - [(\text{area at } t_{24}/\text{area at } t_0) \times 100]$.

## 2.4. Reverse transcription-polymerase chain reaction (qRT-PCR)

MDA-MB-231 cells were cultured for 2 days in standard conditions or exposed to ELF-EMF. Total RNA was extracted with TRIzol (Invitrogen, Thermo Fisher Scientific, Waltham, MA, USA). Total RNA (1 µg) was reversely transcribed into cDNA and quantitative PCR was carried out as previously described [11]. Specific primers amplified the transcripts of the following human genes: cytochrome c oxidase subunit 2 (COX2), cytochrome c oxidase subunit 4 (COX4), adenosine triphosphate (ATP) synthase subunit beta (ATP5B), mitochondrial ATP synthase F0 subunit 6 (MT-ATP6).

## 2.5. Measurement of the mitochondrial membrane potential ($\Delta\Psi$m)

JC-1 (5,5′,6,6′-tetrachloro-1,1′,3,3′-tetraethylbenzimidazolylcarbocyanine iodide), a mitochondrial dye that stains mitochondria in living cells in a membrane potential-dependent fashion, was used to determine $\Delta\Psi$m by flow cytometry, as previously reported [12]. The potential-dependent accumulation in mitochondria is indicated by a fluorescence emission shift from green to red, due to the formation of red fluorescent J-aggregates. Consequently, the enhanced mitochondrial activity is indicated by an increase in the red/green fluorescence intensity ratio (FL2/FL1 channels of flow cytometer). After 2 days of exposure, HT-29 cells were loaded with the dye and analysed as previously described [10].

## 2.6. Cell proliferation assay

A total of 2000 cells were seeded on 96-multiwell plates and cultured for 3 days in the presence or absence of the specific electromagnetic field. At the end of this period, cells were fixed for 15 min with 11% glutaraldehyde; plates were washed three times, air-dried and stained for 20 min with 0.1% crystal violet solution. The bound dye was solubilized with 10% acetic acid solution and the absorbance was determined at 595 nm. Values from 12 wells were averaged for each experimental condition and the experiment was repeated three times.

## 2.7. Statistical analysis

Statistical analysis of data was performed by means of the unpaired two-tailed Student's $t$-test or using the ANOVA test with Tukey's *post hoc* correction, where relevant. $p$-values of less than 0.05 were considered significant. All data were expressed as mean ± s.d. of three independent experiments.

# 3. Results

## 3.1. Thermodynamic analysis

The metabolism of living cells involves flows of matter and heat into and out of the cells; the heat flux is the heat wasted by the cell towards its environment. We can consider that, inside the experimental set-up usually used in the biophysical and biochemical analysis of cells, the general approach to the heat transfer (lumped element model to black box) results in the following equation [13,14]:

$$
\begin{cases}
\dfrac{\partial^2 T_{\text{cell}}}{\partial r^2} - \dfrac{H_M}{\lambda} = \dfrac{1}{a}\dfrac{\partial T_{\text{cell}}}{\partial t} \\
\dfrac{\partial T_{\text{cell}}}{\partial t} = -\dfrac{T_{\text{cell}} - T_0}{\tau}
\end{cases},
\tag{3.1}
$$

where $r$ is a radial variable, considering the cell as a theoretical sphere, $T_{\text{cell}}$ is the temperature, $H_M$ is the metabolism, $a = \lambda/\rho c_{\text{cell}}$, with $\rho$ density and $c_{\text{cell}}$ specific heat, $T_0$ is the environmental temperature, $\tau = \rho c_{\text{cell}} V/\alpha A$, $V$ is the volume and $A$ is the area of the cell, and $\alpha$ is the coefficient of convection. This equation leads to a harmonic solution [14]

$$
T_{\text{cell}}(r) = T_0 + (T_{\text{cell}} - T_0)\,\sin\left(\frac{r}{\sqrt{a\tau}}\right) - \frac{a\tau H_M}{\lambda}.
\tag{3.2}
$$

$H_M$ can also include the mitochondrial metabolic activity producing heat. Some relevant data of the cells are calculated [15] as the mass density $\rho \approx 10^3\ \text{kg m}^{-3}$, the specific heat $c \approx 0.6\ \text{J kg}^{-1}\ \text{K}^{-1}$, $\alpha = 0.023\text{Re}^{0.8}\text{Pr}^{0.35}\lambda/\langle R \rangle$ where $\lambda \approx 0.6\ \text{W m}^{-1}\ \text{K}^{-1}$ is the conductivity, $\text{Re} \approx 0.2$ Reynolds' number and $\text{Pr} \approx 0.7$ Prandt's number. In relation to these data we can evaluate the Biot number $\text{Bi} \approx 0.0056$; therefore, we can study heat transfer by using the lumped element model. We can consider that, inside the experimental set-up usually used in the biophysical and biochemical analysis of cells, heat flux is exchanged by convection with the aqueous solution around any cell, therefore, we can write

$$
\dot{Q} = \rho_{\text{cell}} V c_{\text{cell}} \frac{\text{d}T_{\text{cell}}}{\text{d}t} = \alpha A (T_{\text{cell}} - T_0) = \alpha \frac{V}{\langle R \rangle}(T_{\text{cell}} - T_0),
\tag{3.3}
$$

where $\alpha$ is the coefficient of convection, $A = V/\langle R \rangle$ is the surface area of the cell, which changes with the phases of cell development, $V$ is the volume of the cell, $\langle R \rangle$ is the volume/area ratio, a parameter which influences the chemical reaction time and the fluxes through the cell membrane, $\rho_{\text{cell}}$ is the cell mass density, $c_{\text{cell}}$ is the specific heat of the cell, and $(T_{\text{cell}} - T_0)$ is the difference of temperatures between the cell temperature and the environment temperature. The term $A = V/\langle R \rangle$ is the geometric shape of the cell in relation to convection. We must introduce this quantity because a cell has a definite volume at each stage of its life, but it can change its shape in relation to its duplication phase at the time of observation.

Now, from equation (3.3), we can obtain [13]

$$
\frac{\text{d}\ln(T_{\text{cell}} - T_0)}{\text{d}t} = \frac{\alpha}{\rho_{\text{cell}} c_{\text{cell}}}\frac{1}{\langle R \rangle} = \frac{1}{\tau} = \nu.
\tag{3.4}
$$

It is evident that the greater the volume–area ratio, the lower the thermal exchange, where $\alpha$, $\rho_{cell}$ and $c_{cell}$ are approximately constant. This relationship is important because it means that the cell adapts its volume/area rate in order to optimize the cell membrane fluxes, and to perform the required work and, conversely, this geometric rate also controls heat exchange.

Equation (3.4) is that which links the frequency $v = 1/\tau = (\alpha/\rho_{cell} c_{cell} \langle R \rangle)$ to the structural and geometrical properties of the cell and its environment, in relation to heat exchange. This frequency is obtained by introducing the electric analogy in the heat exchange; thus, $1/\alpha A$ is the thermal resistance, while $\rho_{cell} c_{cell} V$ is the thermal accumulation capacity. Consequently, this relation (equation (3.4)) is the thermal analogy of the resonant frequency for a resistor–capacitor circuit. Equation (3.4) is the relation used for calculating the frequencies of ELF-EMF effective on cells in the experimental validation.

We can define this frequency by our thermodynamic approach. Indeed, each system presents a characteristic response time to the external thermal perturbation. We suggest that this frequency is the inverse of the characteristic response time of the cell to the external thermal perturbation, or the heat exchange rate. Indeed, the heat flow can be written as

$$\dot{Q} = \frac{Q}{\tau} = Qv \tag{3.5}$$

where $Q$ is the heat wasted during the cell life.

Moreover, we can consider that the ion fluxes are controlled by the membrane potential, which is related to the Gibbs free energy, by the relation [14–18]

$$dG = d\phi - 2.3 \frac{RT_0}{F} dpH, \tag{3.6}$$

where $d\phi$ is the variation of electric potential, $R$ is the universal constant of gases and $F$ is the Faraday constant. At the stationary states, it follows

$$d\phi = 2.3 \frac{RT_0}{F} dpH. \tag{3.7}$$

The equation links the variation of the cell membrane electric potential to the variation of the pH, related to the ions fluxes. Conversely, we can try to force a variation of the pH by varying the cell membrane electric potential. We can induce a variation in the cell membrane electric potential by using an electromagnetic field, with a frequency equal to the characteristic frequency of our system.

When we apply a perturbation to a cell by an electromagnetic wave at the calculated thermodynamic frequency, we expect to force the cell state, producing an amplification of the heat exchanged. As previously discussed [8], the interaction between the electromagnetic waves and the cell can be described as a resonance of the biomolecular structures of the cell membrane. In this study, we propose that the electromagnetic field can exert a remarkable effect on resonant intracellular molecular oscillations.

## 3.2. The biochemical model envisages the effects of thermomagnetic resonance

At resonant frequencies, small periodic driving forces have the ability to produce oscillations with large amplitude, due to the storage of vibrational energy. In other words, if we apply the correct stimulus, we can push (amplify) the oscillatory events and thus magnify the heat exchanged. The thermomagnetic effect consists of the entropy variation due to applying a magnetic field. Distinct vibrational modes of proteins in the high-frequency domain result from intra-molecular degrees of freedom of proteins, so that every protein has a specific vibrational signature. Among them, we considered tubulin polymerization, which has been recently investigated and found to be sensitive to particular resonance frequencies of tubulin. The microtubule is, in fact, a vibrating resonant string [19,20], and the recent study by Sahu *et al.* [21] has demonstrated that the electromagnetic wave can synchronize microtubule vibration and enhance filament self-assembly even in the absence of guanosine triphosphate (GTP). Moreover, the coupling of electromagnetic radiation to mechanical vibrations of microtubules has been investigated as an important basis for biological and medical applications [22]. The variation of tubulin polymerization may affect many intracellular events, since microtubules play a central role in a variety of cellular activities, such as mitosis, intracellular trafficking, cell shape and motility; we decided to investigate cell motility, the effects on mitochondrial activity and proliferation.

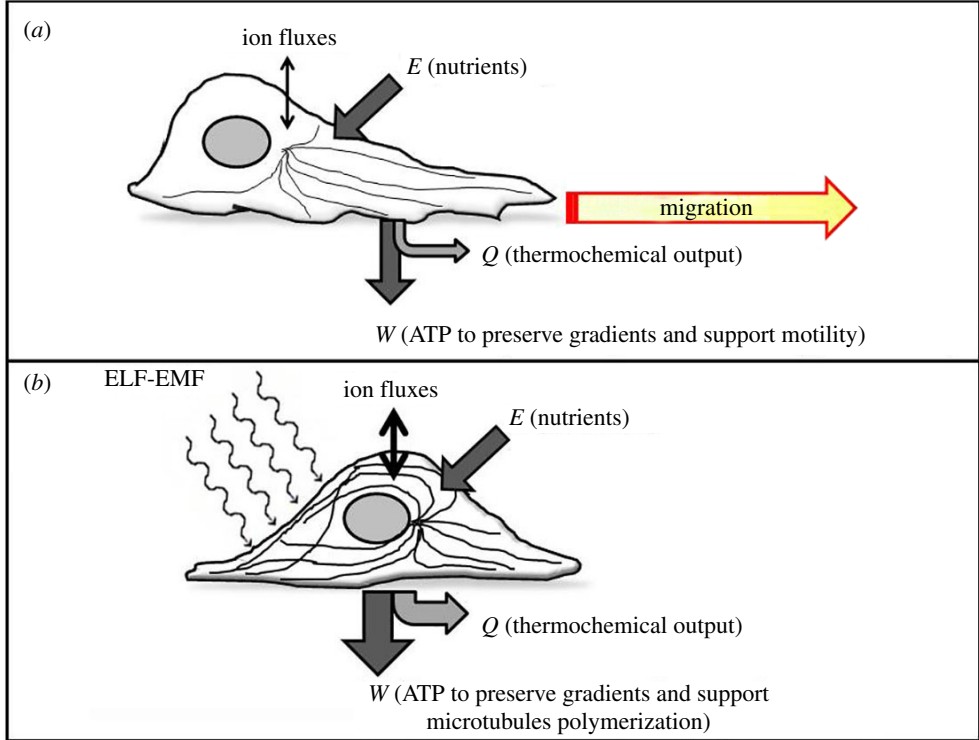

**Figure 1.** ELF-EMF exerts a resonance effect on tubulin polymerization. (a) Transformation of energy (E) from nutrients into work (W) allows cytoskeletal reorganization; tubulin remodelling facilitates movement. Heat production (Q) and entropy generation of the cell–environment system represent the thermochemical output analysed in the environment. (b) When the cell is exposed to a specific resonant ELF-EMF the reinforced polymerization of tubulin filaments makes migration difficult. The increased demand for ATP necessary to preserve gradients is sustained by polymerized tubulin enhancing mitochondrial activity. Electromagnetic stimulation produces the maximum entropy generation of the cell–environment system and increases heat dissipation.

In proliferating cells, microtubules are one of the essential components in the division process, as they form the mitotic spindle. Due to the dynamic nature of microtubules through polymerization and depolymerization cycles, many drugs that stabilize microtubules exert anti-proliferative and anti-cancer activity [23].

The movement of cells is supported by dynamic changes in the cytoskeleton; in addition to the central role played by actin polymerization and by the actin/myosin contractile forces causing cell migration, reorganization of tubulin polymers takes part in the remodelling that polarizes the cell and establishes directionality of the movement. Suppressing microtubule dynamics inhibits the rate of migration [24], and upregulation of tubulin acetylation promotes its polymerization and slows down cell motility [25]; conversely, microtubule destabilization has been reported to facilitate focal adhesion assembly and thereby enhance cell migration [26]. It has been demonstrated that the deacetylation of tubulin decreases microtubule stability and mediates the increase of motility triggered by transforming growth factor (TGF) $\beta$ [25]; this activity is a fundamental step in the epithelial-mesenchymal transition and formation of metastasis [27].

If we apply a resonant electromagnetic wave to a migrating cell, the hypothesized amplified microtubule polymerization will cancel the dynamic remodelling requested to move forward; microtubules will equally pull the rear and the front of the polarized cell, and cell migration will be hindered. Our model suggests that the ELF-EMF could stabilize microtubule structure and reduce cell motility, as shown in figure 1.

Another intriguing effect of the electromagnetic field, at the specific frequency thought to modulate tubulin vibration and microtubule assembly, could be exerted on mitochondrial energy metabolism. Microtubule polymerization is an energy-dependent process, since it is driven by GTP hydrolysis, which is synthesized by mitochondrial activity either directly or from ATP; ATP also promotes polymerization [28,29]. Microtubule formation is indeed linked to mitochondrial activity, as suggested by the study by Sheldon et al. [30] who demonstrated the modulation of the mitochondrial voltage-dependent anion channels (VDAC) permeability to ATP/ADP and to other mitochondrial respiratory

substrates by the changing pool of dimeric tubulin in the cytosol. Based on these observations, we can reasonably suppose that the presumed increase of microtubule polymerization caused by the resonant electromagnetic field would enhance the production of the necessary mitochondrial energy by controlling the exchange of ATP/ADP at the VDAC channel site.

In conclusion, as a result of our biochemical analysis, we propose that the effect of the resonant electromagnetic stimulation could consist of enhanced microtubule assembly, in the loss of movement and in the increase of mitochondrial energy production necessary to sustain microtubule rearrangements, as summarized in figure 1.

## 3.3. Experimental validation

In previous studies, we calculated the frequencies of the electromagnetic wave expected to influence cancer cell biochemistry, and these frequencies have been experimentally verified [8,11]. In fact, the proliferation of several human cancer cell lines was evaluated, and in every experiment, the electromagnetic field was effective in inhibiting the growth of one cell type, as predicted by the mathematical model. In this study, we determined the resonant frequencies specific for two experimental models of human cancer cells, based on their morphology and size; we exposed the cells to ELF-EMF and we assessed the effects of the thermomagnetic resonance. We measured the cell size, we calculated the volume and the area of the cells and then we determined the frequency of the electromagnetic field using equation (3.4). The frequency calculated by the mathematical model as the most effective on each cell type was 6 Hz for MDA-MB-231 cells and 24 Hz for HT-29 cells.

To investigate the biological response hypothesized by our thermodynamic/biochemical model, we measured cell motility as the migration of the two human cancer cell lines in the wound healing assay. After 2 days of ELF-EMF exposure, the MDA-MB-231 breast cancer cells showed a decreased motility compared with untreated cells (figure 2*a*). The experiment was repeated on the colon cancer cell line HT-29, with the same results (figure 2*b*). Interestingly, the effect of TGF $\beta$ on migration was completely reversed by the electromagnetic field (figure 2*b*); as TGF $\beta$ promotes the instability of tubulin filaments in order to enhance cell migration, the inhibition of its activity by the electromagnetic radiation supports the possibility that the electromagnetic field affects microtubule polymerization.

We then evaluated the effects of ELF-EMF on the mitochondrial activity of the two human cancer cell lines. After exposure to the electromagnetic field, we found increased expression of the complexes of the respiratory chain in the breast cancer cells MDA-MB-231 (figure 3*a*) and we observed enhancement of mitochondrial activity in colon cancer cells HT-29, assessed as increased membrane potential (figure 3*b*). As we previously demonstrated [8,11], exposure to specific frequencies of ELF-EMF can decrease cell growth. In this study, we also observed that the MDA-MB-231 cells and HT-29 cells exposed to ELF-EMF reduced their proliferation (figure 3*c*).

# 4. Discussion

The thermodynamic approach is a global approach in oncology because it evaluates cell behaviour by considering only the energy- and mass-flux balances during the entire cycle of cell life. Genetic anomalies are not assessed, instead, their consequences are evaluated as an expression of the energy conversion in the cell. Similarly, the precise molecular mechanisms responsible for the biological effects triggered by the electromagnetic field are not investigated by the thermodynamic method, and the analysis of the control of fluxes (ion and energy fluxes) is carried out by thermodynamic considerations. Many valuable theories could explain the efficacy of the parameters of the ELF-EMF selected by our thermodynamic analysis. Ion cyclotron resonance (ICR) is one among many possible mechanisms that can explain observed interactions between weak low-frequency electromagnetic fields (from a few to several hundred hertz) and biological systems [31]. The ICR hypothesis suggests that the physiological activity of ions involved in cell signalling processes, such as calcium, can be altered when the ratio of applied signal frequency to intensity is equal to the ionic charge-to-mass ratio. For any given ionic species, the specific frequency is called the cyclotronic frequency. Another established theory is based on the forced vibration of the free ions on the surface of a cell membrane, caused by an external oscillating field [18]. Oscillating ions can perturb the gating of channels that are electrically or mechanically sensitive, altering the electrochemical balance of the plasma membrane and, therefore, the entire cell function. Since the amplitude of the forced vibration is found to be inversely proportional to the frequency of the EMF, the low-frequency fields appear to be more bioactive.

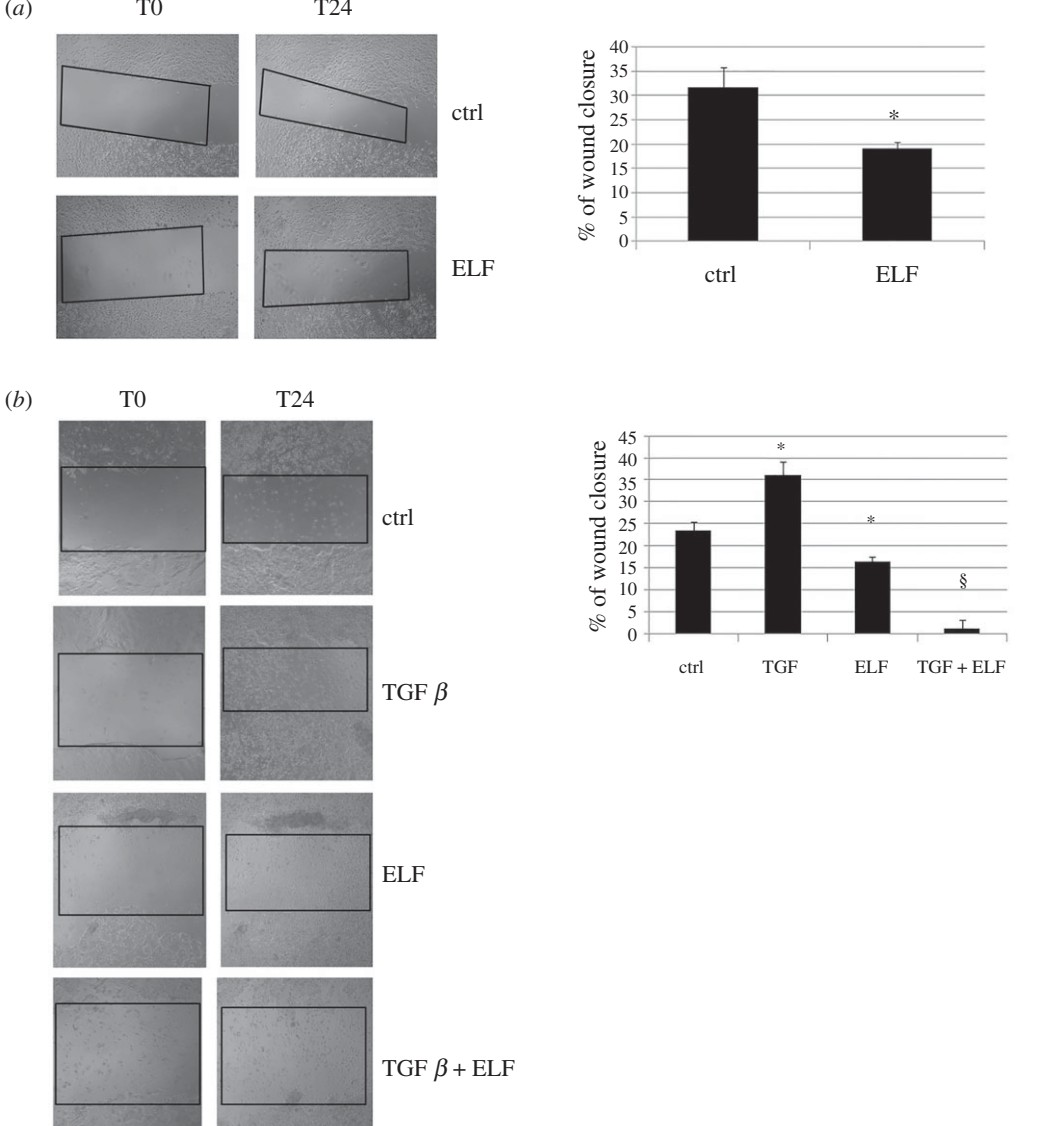

**Figure 2.** ELF-EMF inhibits cell motility. The effects on cell migration at 0 and 24 h were evaluated by a wound-closure assay. Under different experimental conditions, the empty areas in the wound-closure assay were measured and graphically presented as the percentage of wound closure as described in Material and methods. The graphs display the means ± s.d. of three independent experiments. *$p < 0.05$ compared with the untreated cells; §$p < 0.05$ compared with the cells treated with TGF-$\beta$ alone.

Recently, we introduced a new thermophysical approach [8,32–33], starting from some previous considerations [14,34–39], and we also obtained some experimental proofs supporting the present results [8,11,39].

In this study, the conclusions reached by the thermodynamic analysis highlight the role of the volume–area ratio of the cells in relation to the control of fluxes and the heat exchanged in *in vitro* experiments.

Indeed, there is a temperature difference between the inside of a living cell and its environment, which is a thermodynamic requirement for life. Heat is exchanged between the inside and outside of the cell due to this temperature difference. This heat flow contributes to entropy generation in the cell environment, and it is fundamental to decrease the entropy within the cell, a process necessary to sustain cell life. Considering the temperature difference between the environment and cell interior has allowed us to introduce non-equilibrium thermodynamics for analysing cell behaviour. Brock suggested that the stability of thermophilic organisms can be attributed to the membrane structure properties of these organisms [40]. The temperature gradient contributes to the flow of substances

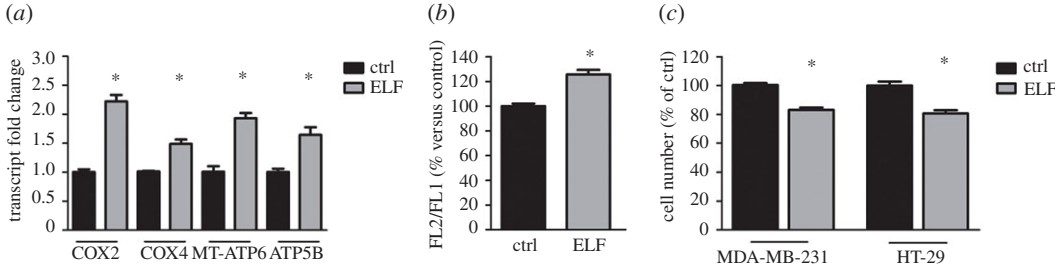

**Figure 3.** Exposure to ELF-EMF increases mitochondrial activity and reduces proliferation. (a) RT-PCR analysis of two subunits of the respiratory chain COX2 and COX4 and two subunits of ATP synthase MT-ATP6 and ATP5B. The expression of transcripts was evaluated in untreated MDA-MB-231 cells (ctrl) and cells treated at the specific frequency. Fold changes versus control are plotted on the graph. (b) The mitochondrial membrane potential of HT-29 cells was examined using JC-1 cytofluorimetric evaluation. The FL-2/FL-1 ratio was calculated and expressed as a percentage of the value obtained for untreated cells. (c) After incubation in the presence or absence of the specific irradiation (6 Hz for MDA-MB-231 and 24 Hz for HT-29) for 3 days,` cells were subjected to crystal violet staining and values were expressed as a percentage of the untreated cells. Data represent the means ± s.d. of three independent experiments. $^*p < 0.05$ compared with the control.

through the cell membranes of the cell, with a consequent influence on metabolic processes [32,33,39]. The approach proposed here has allowed us to evaluate the homeostatic cellular response to external perturbations. This response represents a thermochemical output of the cell in the environment. Therefore, we can suggest that the thermodynamic approach refers to a model of analysis of the action and reaction in terms of membrane flux variation. We have highlighted the existence of a characteristic response time of any cell type to heat exchange. This response time relates to the volume–area ratio of the cell, a geometrical parameter fundamental for considering fluxes, and the electric potential variation of the cell membrane.

The results of our thermodynamic analysis highlight how the irreversibility also plays a fundamental role in biophysical systems; indeed, the geometrical ratio is fully related to the entropy generation as is clear from equation (3.3). This leads to a new approach to biological physics able to evaluate fluxes, based on the first and second laws of thermodynamics, and on the recently introduced constructal approach [13,41–46].

Lastly, we have introduced the resonance as a fundamental phenomenon in the biophysical processes related to cancer growth. In particular, we consider that the effect of the resonant frequency consists of modifying the heat flux, which determines a variation of the Gibbs potential, in turn, generating changes in the membrane potential.

We have experimentally tested the effect of thermodynamic resonance by using electromagnetic waves. We have evaluated the possibility to use mechanical waves, which we verified by some results in the literature [47]. In the past, several molecular mechanisms have been proposed to explain the effects of ELF-EMF; however, the effects and the reported molecular events are numerous, and depend on the cellular model studied and the selected electromagnetic field [48–50]. In recent studies, we have verified the biological response of cancer cells to the resonant frequencies of ELF-EMF. Our previous investigations have demonstrated the specificity of the frequency calculated by the mathematical model; in fact, the proliferation of six human cancer cell lines was evaluated, and we demonstrated that we could calculate and validate the frequencies of the ELF-EMF specifically effective on distinct cell types [8]. In a recent study, the MDA-MB-231 cell line was found responsive only to a frequency of 6 Hz, whereas the same frequency was not effective on the growth of another non-malignant human breast cell line [11]. In the present study, we have developed a novel analysis of fluxes and a biochemical model that proposes the possible impact of ELF-EMF on resonant vibration of tubulin structures. The biochemical analysis suggests that the hypothesized increased polymerization of microtubules could influence cell growth, motility and mitochondrial energy production. Microtubules are electrical polar structures with power supply (around $10^{-14}$ W cm$^{-1}$ per unit length of the microtubule) from hydrolysis of GTP to guanosine diphosphate (GDP); the related energy can excite vibrations. Microtubules lose part of the energy by viscous damping of the surrounding cytosol [51]. Microtubules play a fundamental role both in the organization activities of the living cells and in intracellular transport [52]. Sahu et al. [21] showed the sensitivity of microtubules at a frequency of around 1 MHz, but the global effect of the microtubules synchronization in neurons is obtained at the brain 4–40 Hz signal, lower than five orders of

magnitude [20,21,53]. The effect proposed in neurons is just an amplification of a resonant electromagnetic interaction between external very low-frequency electromagnetic waves and cell microtubules, which also generates the synchronization of microtubules. The biological resonant effect described in neurons can occur in all cells, with different global effects in relation to their specialized functions. In our thermodynamic approach to cancer, we exploited the resonant frequencies of ELF-EMF to affect intracellular events; in analogy with the neuronal model, we hypothesized that external low-frequency waves could trigger a high-frequency resonant effect on microtubules controlling cell behaviour. In all cells, the mitochondrial respiratory chain and oxidative phosphorylation convert the energy obtained from nutrients to a proton-motive force driving ATP synthesis, necessary to transport molecules against a gradient, but also cause dispersion of energy as heat. In this context, it is possible to introduce a new thermodynamic approach: cells can be considered as adaptive thermal engines, which convert metabolic fluxes into useful work with a related heat waste towards the environment due to irreversibility [32,33,39]. Human cells must exchange their waste heat with an environment kept at constant temperature (the human body around them), so if a difference in the metabolism and in the efficiency of the cell system occurs, as in cancer, it becomes difficult for the cells to maintain their optimal temperature required for survival. Our mathematical model has taken into consideration the key role of mitochondrial activity in heat fluxes by introducing the factor $H_M$ in equations (3.1) and (3.2); in addition, the biochemical analysis suggests the molecular basis of the possible link between a reinforced microtubule assembly and the consequent energy demand associated with mitochondrial heat waste.

## 5. Limitations

In this study, we combined the thermodynamic approach with a biochemical model able to hypothesize the biological effects of the thermomagnetic resonant stimulation. We focused our analysis on tubulin polymerization because it has been demonstrated that the electromagnetic wave can synchronize microtubule vibration and enhance filament self-assembly [21]; however, we cannot exclude a similar effect on other cytoskeletal components. The experimental data obtained after exposure of the cells to specific frequencies of the electromagnetic field were in agreement with the biological response predicted by the biochemical model, although the involvement of microtubule rearrangements remains speculative and yet to be demonstrated.

## 6. Conclusion

In this article, we have developed the analysis of a thermodynamic approach to cancer cells, with particular focus on the role of the volume–area ratio in the heat exchange, and the consequences on cancer cell behaviour. Thermodynamic analysis has allowed calculation of the resonant frequency of the electromagnetic field able to trigger a thermomagnetic resonance effect. The biochemical model hypothesized the stabilization of microtubules, which affects proliferation, cell migration and mitochondrial activity. The experimental validation of the thermodynamic theoretical model showed that, by applying electromagnetic fields, cell invasion and proliferation can be decreased.

This novel approach could be exploited in designing an innovative support to present anti-cancer therapies, by introducing variation of external fields, at the proper response time. The encouraging experimental results obtained *in vitro* warrant further investigation *in vivo*.

Data accessibility. The experimental data are provided in the electronic supplementary material. Patents: Italian Patent no. 102016000083775 approved; International Patent application no. PCT/IB2017/054659, 31 July 2017.
Authors' contributions. F.S. conceived the study, designed the study, participated in data analysis, coordinated the study and drafted the manuscript. U.L. conceived the study, designed the study, participated in data analysis and drafted the manuscript. G.G. participated in the design of the study and data analysis. L.B. carried out the molecular laboratory work, participated in data analysis and carried out statistical analyses. A.P. coordinated the study and critically revised the manuscript. All authors gave their final approval for publication and agreed to be held accountable for the work performed therein.
Competing interests. The authors declare that they have no competing interests.
Funding. This research did not receive any specific grants from funding agencies in the public, commercial or non-profit sectors.
Acknowledgement. We thank Dr Radhika Sita Srinivasan, PhD, for English language editing.

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
