## [Reviewer comments · Royal Society Open Science]

Review History

RSOS-200299.R0 (Original submission)

Review form: Reviewer 1

Is the manuscript scientifically sound in its present form?

Yes

Are the interpretations and conclusions justified by the results?

Yes

Is the language acceptable?

Yes

Do you have any ethical concerns with this paper?

No

Have you any concerns about statistical analyses in this paper?

No

Recommendation?

Accept as is

Comments to the Author(s)

The paper provide both some novel insights to cancer integrative understanding and some experimental in vitro preliminary proof of concept for its management.

I recommend, therefore, to publish the paper in order to foster replication of the reported experiments and promote its translational path.

Review form: Reviewer 2**Is the manuscript scientifically sound in its present form?**

No

Are the interpretations and conclusions justified by the results?

No

Is the language acceptable?

No

Do you have any ethical concerns with this paper?

No

Have you any concerns about statistical analyses in this paper?

No

Recommendation?

Major revision is needed (please make suggestions in comments)

Comments to the Author(s)**Summary**

The examination of electromagnetic effects on excitable tissue has over a hundred-year history and is of critical importance today as evidenced by an exponential growth in the literature. Medical devices based on neuromodulation and electroceutical technology can be brought to market at a fraction of the cost and time of drugs, and therefore innovative methods and results are important to distribute to the scientific community.

The authors develop a unique formula for analyzing how electromagnetic radiation affects the entropy of a cell by amplification of a thermally resonant circuit, analogous to the classic electric circuit formula derived from its resistance and capacitance (RC circuit). They propose a thermal analogy to electrical resistance as the inverse of the cell's heat convection times its surface area, and a thermal analogy to electrical capacitance as the product of the cell's mass density times its specific heat capacity and volume. Using these formulae the resonance frequency and its inverse, the time constant of resonance, are linked to the structural and geometric properties of the cell. The formulae take into account the dynamic shape, volume, surface area, and volume/surface area ratio of the cell in different phases, notably mitosis, as in healthy or tumor cell division.

Thus, the electromagnetic energy is amplified by electrically responsive cellular components such as the cell membrane, based on their structural and geometrical properties, causing vibration and transducing the electromagnetic energy into amplified thermal energy in the cell.

The practical application they explore with an in vitro study is using the thermal perturbation to inhibit the motility i.e. invasiveness of tumor cells, which they demonstrate with wound healing assays. This is a valuable result.

The authors speculate that the low frequency electromagnetic field alters the dynamics (polymerization) of microtubules and has related effects on mitochondria. However, neither the theoretical framework or empirical results support these speculations, nor do the authors suggest further experiments to support or falsify them.

To the Authors:

The work is valuable and I suggest it be published with major revision, the revision not necessarily requiring a lot of work. The theoretical framework is unique to my knowledge and very interesting. However I do not see how you connect it directly to effects on intracellular structures such as microtubules or mitochondria.

Can you firm up the theoretical-empirical connection by showing how you derive the 6 Hz and 24 Hz resonant frequencies for MDA-MB-231 and HT-29 cells, respectively (p. 13, lines 19-22), from your formulae with the required parameters?

The empirical results on motility seem sound and are valuable leads to identifying the underlying mechanism. But cell processes are complicated and the roles of various key molecules, such as cytoskeletal ones, or organelles, such as mitochondria, are complex and manifold.

How would you establish that motility effects are due to microtubule polymerization vs. actin effects?

Microtubules affect many cell processes; polymerization alone affects many cell processes – are other observable processes disrupted by the fields? Why or why not?

How would you prove effects on microtubule polymerization? For instance, Giladi et al. showed a 12% increase in free tubulin in a study of 200 KHz AC fields on cells.

Giladi M, Schneiderman RS, Voloshin T, Porat Y, Munster M, Blat R, et al. Mitotic Spindle Disruption by Alternating Electric Fields Leads to Improper Chromosome Segregation and Mitotic Catastrophe in Cancer Cells. *Sci Rep* 2015;5:18046.

Will the method affect normal, non-tumor cell motility?

Will the method affect non-tumor cell mitochondria? Why or why not? If so, what would be the effects?

How do the electromagnetic fields affect mitochondria, specifically? Take for example this sentence (p 18, lines 24-28):

“In this work we link the effect on proliferation and mitochondrial activity to a novel theoretical model that take into account the vibration of microtubules.”

I don't see theoretical or empirical support for that statement in the paper. It is not necessarily surprising that infusing energy into the cell affects mitochondria, which you do demonstrate. But you speculate there may be a connection between microtubule effects, mitochondria, and motility. You could extend the thermodynamic resonance analysis to the mitochondrial membranes, which would be interesting as the strongest electric field in cells is there, and that may refute the Wenger et al. numerical model, which showed that it takes several hundred KHz frequency for appreciable energy to penetrate the cell membrane.

Wenger C, Giladi M, Bomzon Z, Salvador R, Bassar PJ, Miranda PC Modeling Tumor Treating Fields (TTFields) application in single cells during metaphase and telophase. *Conf Proc IEEE Eng Med Biol Soc* 2015;2015:6892-6895.

I suggest you clearly label speculations as such and add a Limitations section in which you list the limitations of the method and experiments. Beginning with the Bandyopadhyay reference, the ideas are so speculative (theirs and yours) I would take them out as essentially unrelated to your presented work.

Readability and Clarity

The paper would be improved for readability and clarity by someone with a fluent command of English taking a pass through fixing grammar, cutting some wordiness, etc. There are too many specific suggestions to list, but by way of example, here is pg 17, lines 33-38, as is, and suggested revision for clarity:

“We have pointed out the existence of a proper time of answer of any cell type to the heat exchange. This time results related to the cells volume-area ratio, a geometrical parameter fundamental for the considerations on the fluxes and cells membrane electric potential variation.”

‘We have pointed out the existence of a characteristic response time of any cell type to heat exchange. This response time r relates to the cell’s volume-area ratio, a geometrical parameter fundamental for consideration of fluxes and the cell’s membrane electric potential variation.’

Minor points

Ref 15 Morowitz, Energy Flow, is a duplicate of Ref. 3.

Pg 16 line 58 typo: heat, not heath

Decision letter (RSOS-200299.R0)

Dear Dr Silvagno,

The editors assigned to your paper ("Thermomagnetic resonance affects cancer growth and motility") have now received comments from reviewers. We would like you to revise your paper in accordance with the referee and Associate Editor suggestions which can be found below (not including confidential reports to the Editor). Please note this decision does not guarantee eventual acceptance.

Please submit a copy of your revised paper before 20-May-2020. Please note that the revision deadline will expire at 00.00am on this date. If we do not hear from you within this time then it will be assumed that the paper has been withdrawn. In exceptional circumstances, extensions may be possible if agreed with the Editorial Office in advance. We do not allow multiple rounds of revision so we urge you to make every effort to fully address all of the comments at this stage. If deemed necessary by the Editors, your manuscript will be sent back to one or more of the original reviewers for assessment. If the original reviewers are not available, we may invite new reviewers.

When submitting your revised manuscript, you must respond to the comments made by the

referees and upload a file "Response to Referees" in "Section 6 - File Upload". Please use this to document how you have responded to the comments, and the adjustments you have made. In order to expedite the processing of the revised manuscript, please be as specific as possible in your response.

- Data accessibility

If you wish to submit your supporting data or code to Dryad (<http://datadryad.org/>), or modify your current submission to dryad, please use the following link:
<http://datadryad.org/submit?journalID=RSOS&manu=RSOS-200299>

- Competing interests

- Authors' contributions

- Acknowledgements

- Funding statement

on behalf of Professor Guy Genin (Associate Editor) and Catrin Pritchard (Subject Editor)
openscience@royalsociety.org

Associate Editor's comments (Professor Guy Genin):
Comments to the Author:

Thank you for sending this paper to RSOS! Like the reviewers, I am enthusiastic about this contribution. Reviewer 2 points out, and I agree, that several instances exist in which statements are made that are not fully supported by the results, including links to certain sub cellular phenomena. In the revision, please be certain to move all such statements to the discussion and make clear to the reader which are supported fully and which are speculation. I look forward to reading the revised paper!

Comments to Author:

Reviewers' Comments to Author:
Reviewer: 1

Comments to the Author(s)
The paper provide both some novel insights to cancer integrative understanding and some experimental in vitro preliminary proof of concept for its management.
I recommend, therefore, to publish the paper in order to foster replication of the reported experiments and promote its translational path.

Reviewer: 2

Comments to the Author(s)
Summary

The examination of electromagnetic effects on excitable tissue has over a hundred-year history and is of critical importance today as evidenced by an exponential growth in the literature. Medical devices based on neuromodulation and electroceutical technology can be brought to market at a fraction of the cost and time of drugs, and therefore innovative methods and results are important to distribute to the scientific community.

The authors develop a unique formula for analyzing how electromagnetic radiation affects the entropy of a cell by amplification of a thermally resonant circuit, analogous to the classic electric circuit formula derived from its resistance and capacitance (RC circuit). They propose a thermal analogy to electrical resistance as the inverse of the cell's heat convection times its surface area, and a thermal analogy to electrical capacitance as the product of the cell's mass density times its specific heat capacity and volume. Using these formulae the resonance frequency and its inverse, the time constant of resonance, are linked to the structural and geometric properties of the cell.

The formulae take into account the dynamic shape, volume, surface area, and volume/surface area ratio of the cell in different phases, notably mitosis, as in healthy or tumor cell division.

Thus, the electromagnetic energy is amplified by electrically responsive cellular components such as the cell membrane, based on their structural and geometrical properties, causing vibration and transducing the electromagnetic energy into amplified thermal energy in the cell.

The practical application they explore with an in vitro study is using the thermal perturbation to inhibit the motility i.e. invasiveness of tumor cells, which they demonstrate with wound healing assays. This is a valuable result.

The authors speculate that the low frequency electromagnetic field alters the dynamics (polymerization) of microtubules and has related effects on mitochondria. However, neither the theoretical framework or empirical results support these speculations, nor do the authors suggest further experiments to support or falsify them.

To the Authors:

The work is valuable and I suggest it be published with major revision, the revision not necessarily requiring a lot of work. The theoretical framework is unique to my knowledge and very interesting. However I do not see how you connect it directly to effects on intracellular structures such as microtubules or mitochondria.

Can you firm up the theoretical-empirical connection by showing how you derive the 6 Hz and 24 Hz resonant frequencies for MDA-MB-231 and HT-29 cells, respectively (p. 13, lines 19-22), from your formulae with the required parameters?

The empirical results on motility seem sound and are valuable leads to identifying the underlying mechanism. But cell processes are complicated and the roles of various key molecules, such as cytoskeletal ones, or organelles, such as mitochondria, are complex and manifold.

How would you establish that motility effects are due to microtubule polymerization vs. actin effects?

Microtubules affect many cell processes; polymerization alone affects many cell processes – are other observable processes disrupted by the fields? Why or why not?

How would you prove effects on microtubule polymerization? For instance, Giladi et al. showed a 12% increase in free tubulin in a study of 200 KHz AC fields on cells.

Giladi M, Schneiderman RS, Voloshin T, Porat Y, Munster M, Blat R, et al. Mitotic Spindle Disruption by Alternating Electric Fields Leads to Improper Chromosome Segregation and Mitotic Catastrophe in Cancer Cells. *Sci Rep* 2015;5:18046.

Will the method affect normal, non-tumor cell motility?

Will the method affect non-tumor cell mitochondria? Why or why not? If so, what would be the effects?

How do the electromagnetic fields affect mitochondria, specifically? Take for example this sentence (p 18, lines 24-28):

“In this work we link the effect on proliferation and mitochondrial activity to a novel theoretical model that take into account the vibration of microtubules.”

I don't see theoretical or empirical support for that statement in the paper. It is not necessarily surprising that infusing energy into the cell affects mitochondria, which you do demonstrate. But you speculate there may be a connection between microtubule effects, mitochondria, and

motility. You could extend the thermodynamic resonance analysis to the mitochondrial membranes, which would be interesting as the strongest electric field in cells is there, and that may refute the Wenger et al. numerical model, which showed that it takes several hundred KHz frequency for appreciable energy to penetrate the cell membrane.

Wenger C, Giladi M, Bomzon Z, Salvador R, Basser PJ, Miranda PC Modeling Tumor Treating Fields (TTFields) application in single cells during metaphase and telophase. Conf Proc IEEE Eng Med Biol Soc 2015;2015:6892-6895.

I suggest you clearly label speculations as such and add a Limitations section in which you list the limitations of the method and experiments. Beginning with the Bandyopadhyay reference, the ideas are so speculative (theirs and yours) I would take them out as essentially unrelated to your presented work.

Readability and Clarity

The paper would be improved for readability and clarity by someone with a fluent command of English taking a pass through fixing grammar, cutting some wordiness, etc. There are too many specific suggestions to list, but by way of example, here is pg 17, lines 33-38, as is, and suggested revision for clarity:

“We have pointed out the existence of a proper time of answer of any cell type to the heat exchange. This time results related to the cells volume-area ratio, a geometrical parameter fundamental for the considerations on the fluxes and cells membrane electric potential variation.”

‘We have pointed out the existence of a characteristic response time of any cell type to heat exchange. This response time r relates to the cell’s volume-area ratio, a geometrical parameter fundamental for consideration of fluxes and the cell’s membrane electric potential variation.’

Minor points

Ref 15 Morowitz, Energy Flow, is a duplicate of Ref. 3.

Pg 16 line 58 typo: heat, not heath

Author's Response to Decision Letter for (RSOS-200299.R0)

See Appendix A.

RSOS-200299.R1 (Revision)

Review form: Reviewer 2

Is the manuscript scientifically sound in its present form?

Yes

Are the interpretations and conclusions justified by the results?

Yes

Is the language acceptable?

Yes

Do you have any ethical concerns with this paper?

No

Have you any concerns about statistical analyses in this paper?

No

Recommendation?

Accept as is

Comments to the Author(s)

Congratulations on producing an interesting and valuable paper with unique results. Dr. Srinivasan did an excellent job improving the English, which adds considerably to the paper. The logic of the hypotheses regarding microtubules and mitochondria is clearer to me now and your caveats and Limitations are prudent. I think there are other interesting applications of this work and hope it gets noticed so that others carry it forward as well as your group.

Decision letter (RSOS-200299.R1)

Dear Dr Silvagno,

It is a pleasure to accept your manuscript entitled "Thermomagnetic resonance affects cancer growth and motility" in its current form for publication in Royal Society Open Science. The comments of the reviewer(s) who reviewed your manuscript are included at the foot of this letter.

on behalf of Professor Guy Genin (Associate Editor) and Catrin Pritchard (Subject Editor)
openscience@royalsociety.org

Subject Editor Comments to Author (Professor Catrin Pritchard):

Comments to the Author:

This is an excellent piece of work and we are delighted to accept!

Associate Editor Comments to Author (Professor Guy Genin):

Comments to the Author:

Congratulations on an outstanding contribution. Many thanks for choosing Royal Society journals for your work!

Reviewer comments to Author:

Reviewer: 2

Comments to the Author(s)

Congratulations on producing an interesting and valuable paper with unique results. Dr. Srinivasan did an excellent job improving the English, which adds considerably to the paper. The logic of the hypotheses regarding microtubules and mitochondria is clearer to me now and your caveats and Limitations are prudent. I think there are other interesting applications of this work and hope it gets noticed so that others carry it forward as well as your group.

Appendix A

Dear Editor,

This letter is to submit a revised version of the manuscript. English language has been revised by a native English speaker. Language corrections are in orange in the revised manuscript with track changes.

The manuscript has been modified in order to address the observations made by the Editor and the Reviewer. We thank the Editor and the Reviewers for their interest and their most thoughtful suggestions to improve this article, which prompted a further analysis of our model and its significance. We extensively revised the manuscript taking into account the Reviewer's comments and we added a new section to clarify the speculative aspects of our work.

We believe that the article has gained consistency, clarity and general relevance; I hope the revised manuscript will be judged suitable for publication in Your Journal.

Yours sincerely,

Francesca Silvagno, Ph.D.

Corresponding Author

Response to Editor and Reviewers:

Associate Editor's comments (Professor Guy Genin):

Comments to the Author:

Thank you for sending this paper to RSOS! Like the reviewers, I am enthusiastic about this contribution. Reviewer 2 points out, and I agree, that several instances exist in which statements are made that are not fully supported by the results, including links to certain sub cellular phenomena. In the revision, please be certain to move all such statements to the discussion and make clear to the reader which are supported fully and which are speculation. I look forward to reading the revised paper!

Author's response: We thank the Editor and Reviewers for their interest and their most thoughtful suggestions to improve this article.

We have extensively revised the manuscript; we eliminated speculative statements from the results, and modified the discussion in order to reinforce hypotheses with data from the literature. We added a section "limitations", as suggested, in order to clarify which part of the study is based on biochemical modeling and yet to be verified.

English language has been revised by a native English speaker.

The comments from the Reviewers are addressed in the revised manuscript, as discussed below.

Comments to Author:

Reviewers' Comments to Author:

Reviewer: 1

Comments to the Author(s)

The paper provide both some novel insights to cancer integrative understanding and some experimental in vitro preliminary proof of concept for its management.

I recommend, therefore, to publish the paper in order to foster replication of the reported experiments and promote its translational path.

Author's response: We thank the reviewer for the supporting comments, and we are delighted at the interest raised by our research.

Reviewer: 2

Comments to the Author(s)

Summary

The examination of electromagnetic effects on excitable tissue has over a hundred-year history and is of critical importance today as evidenced by an exponential growth in the literature. Medical devices based on neuromodulation and electroceutical technology can be brought to market at a fraction of the cost and time of drugs, and therefore innovative methods and results are important to distribute to the scientific community.

The authors develop a unique formula for analyzing how electromagnetic radiation affects the entropy of a cell by amplification of a thermally resonant circuit, analogous to the classic electric circuit formula derived from its resistance and capacitance (RC circuit). They propose a thermal analogy to electrical resistance as the inverse of the cell's heat convection times its surface area, and a thermal analogy to electrical capacitance as the product of the cell's mass density times its specific heat capacity and volume. Using these formulae the resonance frequency and its inverse, the time constant of resonance, are linked to the structural and geometric properties of the cell. The formulae take into account the dynamic shape, volume, surface area, and volume/surface area ratio of the cell in different phases, notably mitosis, as in healthy or tumor cell division.

Thus, the electromagnetic energy is amplified by electrically responsive cellular components such as the cell membrane, based on their structural and geometrical properties, causing vibration and transducing the electromagnetic energy into amplified thermal energy in the cell.

The practical application they explore with an in vitro study is using the thermal perturbation to inhibit the motility i.e. invasiveness of tumor cells, which they demonstrate with wound healing assays. This is a valuable result.

The authors speculate that the low frequency electromagnetic field alters the dynamics (polymerization) of microtubules and has related effects on mitochondria. However, neither the theoretical framework or empirical results support these speculations, nor do the authors suggest further experiments to support or falsify them.

To the Authors:

The work is valuable and I suggest it be published with major revision, the revision not necessarily requiring a lot of work. The theoretical framework is unique to my knowledge and very interesting. However I do not see how you connect it directly to effects on intracellular structures such as microtubules or mitochondria.

Author's response: We have used a lumped element model to black box. Consequently, we can take into account the internal cell processes in the H_M term of the new Equation (1), without being

able to highlight the single process. The impact of the electromagnetic field on mitochondria was proposed, and experimentally demonstrated in our previous studies (references 9,11), and in this work the connection is also justified because the internal processes are related to metabolism, biochemical reactions, heat generation, etc. We have included equations that demonstrate the presence of the metabolic heat in the mathematical model; in fact the term H_M in equations (1) and (2) can include the mitochondrial metabolic activity producing heat. We have introduced the involvement of microtubules as a biochemical hypothesis, suggested by the data of other studies, demonstrating that the electromagnetic wave can synchronize microtubule vibrations and enhance filament self-assembly (reference 21); in our biochemical model we proposed the molecular events that could link microtubules with mitochondrial activity. We highlighted the speculative nature of the connection with microtubules in the new section “Limitations”.

Can you firm up the theoretical-empirical connection by showing how you derive the 6 Hz and 24 Hz resonant frequencies for MDA-MB-231 and HT-29 cells, respectively (p. 13, lines 19-22), from your formulae with the required parameters?

Author’s response: we have added the analytical equations in order to make the paper clearer. We hope that it is now more evident how the numerical values were obtained. In particular, from the measured cell size (reported in supplementary material) we calculated the volume of the cells and we determined the area, obtaining the volume/area ratio. We introduced this value into the following relation $\nu = \frac{\alpha}{\rho_{cell} C_{cell}} \frac{1}{\langle R \rangle}$ shown in equation (4), obtaining the frequency. We added a statement in the text (experimental validation, page 14 of the revised text).

The empirical results on motility seem sound and are valuable leads to identifying the underlying mechanism. But cell processes are complicated and the roles of various key molecules, such as cytoskeletal ones, or organelles, such as mitochondria, are complex and manifold.

How would you establish that motility effects are due to microtubule polymerization vs. actin effects?

Author’s response: Of course the electromagnetic field at the applied frequencies could also affect actin polymerization or distribution, which will be experimentally investigated in future studies. We firstly focused our analysis on tubulin because it has been demonstrated that the electromagnetic wave can synchronize microtubule vibrations and enhance filament self-assembly (reference 21). The possible impact on other cytoskeletal structures has been mentioned in the section “Limitations”

Microtubules affect many cell processes; polymerization alone affects many cell processes – are other observable processes disrupted by the fields? Why or why not?

Author’s response: We acknowledge the fact that polymerization affects many processes, and we chose to investigate the events for which we had a biochemical explanation concerning the link with tubulin polymerization, namely cell motility and mitochondrial respiration. On page 11 of the revised text we have stated: “The variation in tubulin polymerization may affect many intracellular events, since microtubules play a central role in a variety of cellular activities, such as mitosis, intracellular trafficking, cell shape and motility”.

Proliferation is among the many processes affected by microtubule polymerization.

During cell proliferation, the microtubule network undergoes a dynamic reorganization, such as the assembly of the mitotic spindle. Suppression of microtubule dynamics not only affects cell motility,

as we suggested, but also proliferation (reference 23). Therefore, the hypothesized effect of the electromagnetic field on tubulin polymerization should not only decrease cell motility but also cell proliferation. Indeed, this effect has been experimentally demonstrated (Figure 3C). We discussed this additional consequence of the hypothesized increase in tubulin polymerization triggered by ELF-EMF on page 11 of the revised text.

How would you prove effects on microtubule polymerization? For instance, Giladi et al. showed a 12% increase in free tubulin in a study of 200 KHz AC fields on cells.

Giladi M, Schneiderman RS, Voloshin T, Porat Y, Munster M, Blat R, et al. Mitotic Spindle Disruption by Alternating Electric Fields Leads to Improper Chromosome Segregation and Mitotic Catastrophe in Cancer Cells. *Sci Rep* 2015;5:18046.

Author's response: we believe that the hypothesized increase of tubulin polymerization would be difficult to demonstrate. Giladi showed that exposure to intermediate-frequency (100–300 kHz) alternating electric fields abated polymerization and mitotic spindle formation. This evident decrease was demonstrated by traditional techniques such as western blotting and fluorescent and immunofluorescent staining. In our experiments, we did not detect multinucleation in replicating cells or differences in cell shapes between treated and control cells, therefore we can exclude the disruption of tubulin polymerization; conversely, the increase of polymerization could be difficult to investigate using the same techniques. More sophisticated microscopy studies on increased polymerization have been performed by the research group of Bandyopadhyay, but we are not able to carry out these experiments.

Will the method affect normal, non-tumor cell motility?

Will the method affect non-tumor cell mitochondria? Why or why not? If so, what would be the effects?

Author's response: In our previous studies, we demonstrated that our thermodynamic approach can calculate specific frequencies for each type of cell, both cancer and non-cancer cells (references 11,39). The specific frequency is calculated on the basis of cell shape and size, as shown by equation (4). Based on our previous work, we expect that the frequencies specific for the cell lines analyzed in this work will not affect other healthy cells, and will not determine the hypothesized resonant effect on tubulin, mitochondria and cell migration of other cells.

How do the electromagnetic fields affect mitochondria, specifically? Take for example this sentence (p 18, lines 24-28):

“In this work we link the effect on proliferation and mitochondrial activity to a novel theoretical model that take into account the vibration of microtubules.”

I don't see theoretical or empirical support for that statement in the paper. It is not necessarily surprising that infusing energy into the cell affects mitochondria, which you do demonstrate. But you speculate there may be a connection between microtubule effects, mitochondria, and motility. You could extend the thermodynamic resonance analysis to the mitochondrial membranes, which would be interesting as the strongest electric field in cells is there, and that may refute the Wenger et al. numerical model, which showed that it takes several hundred KHz frequency for appreciable energy to penetrate the cell membrane.

Wenger C, Giladi M, Bomzon Z, Salvador R, Bassar PJ, Miranda PC Modeling Tumor Treating Fields (TTFields) application in single cells during metaphase and telophase. *Conf Proc IEEE Eng Med Biol Soc* 2015;2015:6892-6895.

Author's response: as a consequence of the lumped system approach, we cannot extend our analytical approach to mitochondria, because the lumped system approach considers the cell as a uniform system. Indeed, our approach takes into account the heat exchange between cells and their

environment, where the cooling fluxes are treated as forced ones. Inside the cell, the fluxes are natural. Evaluation of the convection coefficient depends on the convection process (forced or natural), so the heat exchange evaluation is different between the entire cell and the mitochondria. Moreover, the approach considers the life cycle of the cell, which implies considering the whole cell.

In this study, we combined the thermodynamic approach with a biochemical model able to hypothesize the biological effects of a thermomagnetic resonant stimulation. We admit that the link between the biochemical model and the experimental validation is speculative, we probably did not make this clear enough, and we thank the Reviewer for the valuable comments. We added a section “Limitations” and we modified the above-mentioned statement on page 19 of the revised text.

I suggest you clearly label speculations as such and add a Limitations section in which you list the limitations of the method and experiments. Beginning with the Bandyopadhyay reference, the ideas are so speculative (theirs and yours) I would take them out as essentially unrelated to your presented work.

Author’s response: we take on-board this criticism by the Reviewer, and the suggested section has been added; in fact the theoretical part of the article was already separated from the experimental validation, but in the discussion our speculations were not clearly labeled as such, and we think that the new section improves the scientific value of the paper.

The most speculative ideas from the Bandyopadhyay reference were eliminated from the discussion, and we referred to this author only to discuss an example of biological resonant effect, analogous to the resonant effect proposed by our model in cancer cells.

Readability and Clarity

The paper would be improved for readability and clarity by someone with a fluent command of English taking a pass through fixing grammar, cutting some wordiness, etc. There are too many specific suggestions to list, but by way of example, here is pg 17, lines 33-38, as is, and suggested revision for clarity:

“We have pointed out the existence of a proper time of answer of any cell type to the heat exchange. This time results related to the cells volume-area ratio, a geometrical parameter fundamental for the considerations on the fluxes and cells membrane electric potential variation.”

‘We have pointed out the existence of a characteristic response time of any cell type to heat exchange. This response time r elates to the cell’s volume-area ratio, a geometrical parameter fundamental for consideration of fluxes and the cell’s membrane electric potential variation.’

Minor points

Ref 15 Morowitz, Energy Flow, is a duplicate of Ref. 3.

Pg 16 line 58 typo: heat, not heath

Author’s response: We have revised the entire manuscript as suggested.